# Poultry Meat Quality in Antibiotic Free Production Has Improved by Natural Extract Supplement

**DOI:** 10.3390/ani12192599

**Published:** 2022-09-28

**Authors:** Raffaella Rossi, Francesco Vizzarri, Sabrina Ratti, Carlo Corino

**Affiliations:** 1Department of Veterinary Medicine and Animal Science, Università Degli Studi di Milano, Via Dell’Università 6, 26900 Lodi, Italy; 2Department of Agricultural and Environmental Science, University of Bari Aldo Moro, Via G. Amendola, 165/A, 70126 Bari, Italy; 3Independent Researcher, 20100 Milano, Italy

**Keywords:** growth performances, Hubbard slow growth chicken, polyphenols, seaweed, meat quality

## Abstract

**Simple Summary:**

Farm sustainability is a key factor in animal production. In the recent years, the consumers’ demand for products of animal origin coming from production chains certified for animal welfare has increased. Moreover, the institutions have limited the antibiotic use to prevent the antibiotic resistance. For these reasons, antibiotic-free production chains are emerging. The search for sustainable nutritional approaches able to strengthen animal health and enhance product quality is essential. Natural extracts and seaweed contain several bioactive compounds capable of enhancing animal health and product quality. The present study investigates the effect of dietary supplementation with polyphenols and seaweed mixtures on meat quality parameters in Hubbard slow growth chicken in antibiotic-free production. The present data suggest that dietary supplementation with polyphenols and seaweed mixture increased breast muscle protein content and protect the muscle from oxidative processes, enhancing poultry meat quality parameters.

**Abstract:**

Modern consumers are conscious of the relationship between meat quality and animal welfare. Today, slow-growing chickens are associated with a higher broiler welfare. The present work aims to evaluate the effect of dietary natural extract supplementation with polyphenols and seaweed mixtures (PPE) on breast and thigh muscles quality parameters in Hubbard JA57 slow growth chicken in antibiotic-free production. Five hundred Hubbard female chickens (250 from control and 250 from experimental group) were housed on floor pens (10 pens/treatment, 25 birds/pen) and assigned to two experimental groups: a control group (CON) receiving a commercial diet and another group receiving the same diet supplemented with 0.3% of polyphenols and seaweed mixtures (PPE). Dietary supplementation with PPE did not affect (*p* > 0.05) growth performances. The breast pH tended to be lower (*p* = 0.062) in PPE groups. The protein content of breast muscles resulted higher in PPE samples (*p* < 0.05) than controls. The thigh muscles from PPE group showed a lower (*p* < 0.05) malondialdehyde content than CON during refrigerated storage. In conclusion, the PPE supplement improves breast muscle protein content and oxidative stability of thigh muscle. This feeding practice is suggested to enhance the nutritional and technological parameters of meat Hubbard slow growth chicken in antibiotic-free production.

## 1. Introduction

Recently in poultry production the attention is focused on increasing animal welfare and meat quality. In fact, the food industry should offer alternative products to the consumers, with better impact on their animal welfare and sustainability concept. In the poultry industry, heritage breeds of chickens with slow growth are today associated with higher broiler welfare [1]. So, alternative productions with slow-growing genotypes are emerging.

To improve production efficiency, the broiler selection is focused on animals that grow faster and have heavier slaughter weights than the last decades [2]. In fact, poultry breeds can be classified into fast-growing and slow-growing chickens. Even if there is not a standard classification, the chickens can be categorized based on their average daily gain in fast (>50 g/day) and slow (<50 g/day) growers [3]. Some studies report that fast-growing chickens presented several welfare and health problems due to the rapid growth, high body weight, leg damage and inactivity [4]. In fact, as reported by Bokkers and Koene [5], fast-growing chickens only move to feed and drink after eight weeks of age. Moreover, in fast-growing chickens, a high blood lysozyme level was observed, suggesting acute and chronic inflammation [6].

Meat nutritional characteristics are key factors for a proper food choice and a healthier diet [7]. Chicken meat is low in fat and cholesterol with high digestible proteins and low levels of collagen, and is usually considered healthier than other animal protein sources, especially the red meats [8]. Chicken meat fatty acid composition is also interesting from a nutritional point of view; in fact, it contains a significantly higher amount of monounsaturated fatty acids than bovine or pig meat, and a considerably higher amount of polyunsaturated fatty acids, including omega-6 fatty acids [8].

Moreover, consumers are focused on poultry meat due to ready-to-cook packaging, nutritional and sensory characteristics and lower price compared to pork and beef meat [9]. For these reasons, poultry meat consumption increased in the human diet, and it is estimated to further increase globally from 2021 to 2030, accounting for 52% of the additional meat consumed [10].

Several studies reported that poultry dietary supplementation with seaweed or natural antioxidant improved health and meat quality parameters [11,12].

The nutritional value attributed to seaweeds makes them particularly suitable to be used in livestock animal feed as nutraceuticals for their health benefits, including the prevention of some diseases [13]. The seaweed’s content of several bioactive molecules such as sulfated polysaccharides, phlorotannin, diterpenes, minerals and vitamins makes it a functional dietary ingredient due to its several effects on animal health [14]. Moreover, dietary plant polyphenols could improve growth performances and meat oxidative stability in chicken [15].

Recently, natural mixtures containing brown seaweed and plant polyphenols have been investigated as feed supplements for rabbits to improve health and enhance meat quality traits due to their effects on gut health, feed digestibility, as well as their antioxidant properties [16,17].

In the available literature, no previous study reported the effects of supplementation in poultry with brown seaweed and plant polyphenols mixture, therefore the present study was performed to assess the effect of dietary supplementation with polyphenols and seaweed mixtures (PPE) on meat quality parameters in Hubbard slow growth chicken in antibiotic-free production.

## 2. Materials and Methods

### 2.1. Animals and Diets 

The animals used in this experiment were reared following the European Union guidelines (2010/63/EU) and approved by the Italian Ministry of Health (D. Lgs. n. 26/2014).

The feeding trial was conducted on a commercial farm in the south of Italy (Apulia region, Italy) on 500 Hubbard JA57 female chicks with an average initial weight of 46.5 ± 3.14 g, which were housed on 1.8 m^2^ floor pens (10 pens/treatment, 25 birds/pen) with a stocking density of 14 bird/m^2^. The chickens were housed in a hall raised to the ground under 22–24 °C normal conditions of temperature, with 60–70% humidity and 23 h light regimen (50 Lux in the first week; 30 Lux from the second to eighth week of experiment) throughout the experimental period according to the Hubbard CLASSIC Management Guide. They had free access to the feed and water. 

The chickens were assigned to two experimental groups: a control group (CON) receiving a commercial diet and an experimental group (PPE) receiving the same diet supplemented with 0.3% of mixture containing prebiotic polysaccharides from brown seaweeds (*Laminaria Digitate* and *Hyperborea*, ratio 1:1) plus phenolic acid, hydroxycinnamic acids, tannins, and flavonoids from plant extract (*Castanea sativa*). The phenolic compounds of the supplement were analysed by HPLC-UV–DAD, according to Russo et al. [18], and the beta-carotene quantification was performed in accordance with Rakusa et al. [19]. The chemical composition, analysed according to the methods of the Association of Analytical Chemists [20], detected carotenoid content and the polyphenols composition content of the feed supplement, as reported in Table 1. The basal diet consisted of corn, wheat, soybean meal, sunflower meal, corn gluten and soybean oil formulated according to the requirements of the National Research Council (NRC) [21]. A starter diet containing 21% CP and 13.05 MJ/kg metabolizable energy (ME) was fed from 1 to 28 d of age and a grower diet containing 19% CP and 13.05 MJ/kg ME was fed from 29 to 56 d of age. 

The feeding trial lasted 56 days and the following parameters were monitored throughout the experimental period (1–56 days): initial weight (g), final weight (g) using a Kern DE35K5D (Kern & Sohn GmbH, Balingen, Germany) balance, and feed conversion ratio (g feed/g gain).

### 2.2. Sampling 

The animals (1 chickens/pen/treatment) were slaughtered at 56 days of age at an average weight of 2.09 ± 0.12 kg in a commercial slaughterhouse. Automated equipment was used for stunning, scalding, picking, vent opening, and evisceration. Birds were electrically stunned (11 V, 11 mA, 10 s) and soft-scalded at 53 °C for 120 s. Carcasses were prechilled at 12 °C for 15 min and chilled at 1 °C for 1 h. After chilling, the carcasses were aged on ice for an additional 2.5 h before deboning 4 h postmortem. Ten carcasses per treatment were randomly selected and the breast and thighs were collected, vacuum-packaged and stored at 4 °C. The ice-cooled samples were transported to the lab of the Department of Veterinary Medicine and Animal Science for the determination of meat quality parameters. 

### 2.3. Physical Parameters

All the analyses were performed on right breast muscle (*m. pectoralis major*) and thigh at 0, 3 and 6 days of refrigerated storage. The pH test was performed using a portable pH meter equipped with a meat-penetrating probe (HI98191 microcomputer; Hanna Instruments, Vila do Conde, Portugal), and calibrated with a standard buffer of pH 4.0 and 7.0. The pH value of the breast muscle was performed by inserting a probe electrode into the cranial ventral part of the muscle. In the thigh muscle, the pH value was obtained by inserting a probe electrode at the level of *Biceps femoris*. The pH analysis was carried out with a penetration electrode at three different points of the chicken muscles. 

The colour indexes, lightness (L*), redness (a*), and yellowness (b*), were measured using a CR-300 Chroma Meter (Minolta Camera, Co., Osaka, Japan). The instrument was calibrated using a white calibration plate (Calibration Plate CR-A43; Minolta Camera, Co., Osaka, Japan). The colourimeter had an 8-mm measuring area and was illuminated with a pulsed Xenon arc lamp (illuminat C) at a viewing angle of 0°. Reflectance measurements were obtained at a viewing angle of 0° and the spectral component was included. Each datum is the mean of six replications at the meat sample surface.

### 2.4. Chemical Parameters

The chemical composition of right breast muscle and thigh were determined according to the methods of the Association of Analytical Chemists [21]. Determinations of moisture (method 985.41), ash (method 920.153), fat (method 180 960.39) and crude protein (method 928.08) content were performed in duplicate.

### 2.5. Oxidative Stability

Lipid oxidation in relation to storage time (0, 3 and 6 days) at 4 °C was determined by the thiobarbituric acid reactive substances (TBARS) method of Jo and Ahn [22]. All the analyses were performed in duplicate. The absorbance at 532 nm was measured with Varian Cary 100 UV-VIS spectrophotometer (Varian, Australia). The TBARS value, expressed as the mg malonaldehyde/kg meat, was obtained using a conversion factor based on a standard curve using 1,1,3,3-tetraethoxypropane (TEP; Sigma-Aldrich, Milan, Italy).

### 2.6. Statistical Analyses

Statistical analyses of the data were performed using SPSS (SPSS/24 PC Statistics 26.0 IBM, Armonk, NY, USA). The data on growth performances and meat quality parameters were analysed taking into consideration diet as the main effect. Mortality rates were analysed using Chi-square test. The data on pH, colour parameters and oxidative stability were analysed by repeated measure ANOVA to assess the main effect of treatment and time and their interaction. The data on physical and chemical parameters were analysed taking into consideration muscle as the main effect. Means were compared using Student’s *t* test. Pen was considered as experimental unit for growth performances. Individual bird was considered as experimental unit for meat quality parameters. Data are presented as means ± SEM, and a value of *p* < 0.05 was used to indicate statistical significance.

## 3. Results

### 3.1. Growth Performance 

No effect of dietary treatment (*p* > 0.05) was observed on growth performances. The ADG was unaffected by dietary treatments (35.97 ± 2.51 CON vs. 37.30 ± 3.4 PPE g/day, respectively). The final weight was 2.02 ± 0.49 kg vs. 2.10 ± 0.27 kg in control and PPE group, respectively. In addition, the feed conversion ratio was unaffected (*p* > 0.05) by dietary treatment (2.32 ± 0.50 g/g CON and 2.22 ± 0.54 g/g PPE group). The mortality rates in the two experimental groups displayed a similar value (2.4% CON vs. 2.8% PPE group; *p* > 0.05). 

### 3.2. Physical Parameters

The pH values of breast (*pectoralis major*) muscle and thigh in relation to dietary treatments and storage time are presented in Figure 1.

The pH value of the chicken breast (*pectoralis major*) samples did not change during refrigerated storage and remained stable. Dietary supplementation with PPE tended to lower (*p* = 0.062) pH values during refrigerated storage. In thigh an increase in pH values during refrigerated storage was observed (*p* = 0.001). No difference was observed in relation to dietary treatments. The pH values at different sampling times resulted higher in thigh than breast muscle (6.2 vs. 5.7 day 0; 6.2 vs. 5.8 day 3; 6.6 vs. 5.8 *p* < 0.001).

The changes of breast (*pectoralis major*) muscle and thigh colour indices, lightness (L*), redness (a*) and yellowness (b*) in relation to dietary treatments and storage time at 4 °C are presented in Figure 2A–C, respectively.

The lightness values (L*) of breast (*pectoralis major*) muscle and thigh were significantly affected by storage time (*p* < 0.001) but not by dietary treatment (*p* > 0.05). No interaction between treatment and time was observed (*p* > 0.05). The L* values at the different sampling time displayed higher in breast muscle than in thigh (54.9 vs. 48.3 day 0; 53.6 vs. 46.4 day 3; 51.0 vs. 45.1; *p* < 0.001).

The redness values (a*) were not affected by storage time (*p* > 0.05) and dietary treatment in both samples. No treatment effect or interaction between time and treatment was observed (*p* > 0.05). The a* values at different sampling times displayed higher in thigh than in breast muscle (11.4 vs. 4.3 day 0; 10.9 vs. 3.9 day 3; 10.4 vs. 4.1; *p* < 0.001).

The yellowness (b*) values of breast muscle and thigh were significantly affected by storage time (*p* < 0.001). No treatment effect or interaction between time and treatment was observed (*p* > 0.05). No differences between muscles were observed for b* values at different sampling times (9.0 vs. 7.6 day 0; 12.0 vs. 10.8 day 3; 12.8 vs. 11.5 in thigh and breast, respectively; *p* = 0.100).

### 3.3. Chemical Parameters 

The chemical composition of breast (*pectoralis major*) muscle is reported in Table 2. The chemical composition of breast muscle did not differ (*p* > 0.05) for dry matter, fat content and ash. The crude protein resulted higher (*p* < 0.05) in PPE groups than controls.

The chemical composition of thigh is reported in Table 3. The chemical composition of thigh muscle did not differ (*p* > 0.05) for dry matter, protein, fat content and ash. 

As expected, a different chemical composition was observed between the two muscles considered. A higher content of crude fat was observed in thigh than breast muscle (3.25% vs. 0.94%; *p* < 0.001). The protein content presents an opposite trend and resulted higher in breast than thigh (23.92% vs. 22.5%; *p* < 0.001).

### 3.4. Oxidative Stability

The oxidative stability of the breast (*pectoralis major*) muscle of Hubbard chickens in relation to dietary treatments and storage time is reported in Figure 3. The TBARS values were not affected (*p* > 0.05) by dietary treatment. The time of storage (from 0 d to 6 d) determined a significant increase (*p* < 0.001) in the malondialdehyde (MDA) content in the samples. The TBARS content in muscle of CON group increased from 0.31 mg/kg (initial value) to 0.91 mg/kg at 6 d of refrigerated storage. In the samples from PPE group, the TBARS content increased from 0.27 mg/kg (initial value) to 0.89 mg/kg at 6 d of refrigerated storage. No significant interaction between storage time and dietary treatment was observed (*p* > 0.05).

The oxidative stability of thigh muscle of Hubbard chickens in relation to dietary treatments and storage time is reported in Figure 4. The TBARS values were significantly affected (*p* < 0.05) by dietary treatment. The time of storage (from 0 d to 6 d) determined a significant increase (*p* < 0.001) in the malondialdehyde (MDA) content in all the samples. The TBARS content in muscle of CON group increase from 0.35 mg/kg (initial value) to 1.48 mg/kg at 6 d of refrigerated storage. In the samples from PPE group the TBARS content increased from 0.25 mg/kg (initial value) to 1.04 mg/kg at 6 d of refrigerated storage. No significant interaction between storage time and dietary treatment was observed (*p* > 0.05).

## 4. Discussion

In the present study, the growth performance of the Hubbard chicken was not affected by dietary treatment. No previous study reported the effects of dietary supplementation with prebiotic polysaccharides from brown seaweeds and polyphenols from plant extract in slow growth Hubbard chicken. 

Similar data were reported by Abudabos et al. [23] who observed that the dietary inclusion of Ulva lactuca (30 g/kg) in broiler chicken had no effect on feed intake and body weight gain. In addition, Matshogo et al. [24] reported that green seaweed meal inclusion in Cobb 500 broiler diets had no effects on overall growth performances. It is possible that in chickens, the presence of non-starch polysaccharides such as cellulose and hemicellulose in seaweeds affect digestibility with no effects on growth performance [24]. In disagreement with our data, Choi et al. [25] reported that dietary supplementation with 0.5% of brown seaweed by-products resulted in higher average daily gain in broiler. 

Some studies reported the effects of dietary polyphenols on growth performance in chickens. Dietary supplementation with 20 g/kg grape seed increased body weight and average daily gain in Cobb-500 chicken [26]. Other experimental studies reported a reduction in growth performance with the use of grape seed extract [27]. The different results can be related to the disparate effect of polyphenols on the absorption of nutrients that differ in relation of the type of compounds, its dosage, and the combination with other molecules [28]. 

Meat quality parameters of both breast (*pectoralis major*) muscle and thigh were evaluated. The present data are in line with the pH mean values for breast and thigh chicken muscles, revealing normal meat in all samples [29]. 

Dietary supplementation with PPE tended to lower (*p* = 0.062) the pH values in breast during refrigerated storage. The same results in muscle pH were observed in chicken fed spice extracts [30]. Our previous study in rabbit fed the same feed additive showed that pH parameters at 24 h in *Longissimus lumborum* and *Semimembranosus* muscles were not affected by dietary treatment [17]. The pH values displayed higher in thigh than breast muscle, and in thigh an increase in pH values during refrigerated storage was observed. These results are in line with Sampaio et al. [31], who reported that thigh samples had significantly higher pH values than breast samples, and showed pH value of 5.7–6.4 and 6.3–6.9 during storage, respectively.

The colour parameters of both muscles in the present study were not affected by dietary natural extract supplement in agreement with our previous study in rabbit fed the same mixture [17]. As expected, a high a* value between breast (type IIB white fiber) and thigh (type I and IIA red fiber) muscles due to the level of myoglobin was observed [32]. The lightness (L*) and yellowness (b*) values of breast muscle and thigh were significantly affected by storage time. These data agree with previous study that reported a decrease in L* values and an increase in b* values in both muscles during refrigerated storage [33]. These results showed that the dietary supplementation with polyphenols and brown seaweed mixture had no effect on meat physical parameters.

The dietary supplementation with prebiotic polysaccharides from brown seaweeds and polyphenols from plant extract positively affects nutrition composition of the breast muscle. In fact, an increase in crude protein content was observed in muscle from PPE groups compared to controls. The other nutritional parameters were unaffected by dietary treatment. The same results were observed in chicken fed dietary probiotics and *Yucca schidigera extract*, even if the mechanism of action is still unclear [34]. No effects on meat nutritional composition were observed in rabbits fed the same dietary supplementation [17]. Different nutritional composition was observed in breast and thigh samples, according to previous studies [34,35].

Lipid oxidation is one of the main causes of chicken meat spoilage, decreasing nutritional value due to oxidation of some fatty acids. It is also a health risk for consumers due to the accumulation of oxidation products [36]. The TBARS levels in the breast and thigh meat both shared the same trend but have a higher value in thigh than breast muscle due to the higher fat content. The antioxidant effect of the dietary supplementation with brown seaweeds and plant polyphenols mixture was able to counteract the lipid oxidation, resulting in a better oxidative stability of the thigh meat from PPE experimental group. These results correspond with our previous study in rabbit fed the same mixture [17]. Additionally, lipid oxidation has adverse effects on meat sensory parameters such as texture, flavour, and colour [36]. Study of Zhang et al. [37] reported an improvement in antioxidant activity of poultry meat after dietary supplementation with resveratrol. Moreover, dietary supplementation with pomaces, containing polyphenols, enhance oxidative stability in turkey meat [38]. In addition, Fellenberg et al. [39] indicated that polyphenols from *Quillaja saponaria* protect broiler meat from lipid oxidation.

## 5. Conclusions

In conclusion, dietary supplementation with brown seaweeds and plant polyphenols mixture showed positive effects on meat nutritional parameters. No effects on growth performance of slow-growing Hubbard chickens were observed. Physiochemical parameters were not affected by dietary treatments. The dietary supplement increased breast muscle protein content and improved oxidative stability of thigh muscles. This feeding practice is suggested to enhance the nutritional and technological parameters of meat Hubbard slow growth chicken in antibiotic-free production. Further studies are required to confirm the present data and to better understand the mechanism of action of the active principles from this sustainable supplement.

## Figures and Tables

**Figure 1 animals-12-02599-f001:**
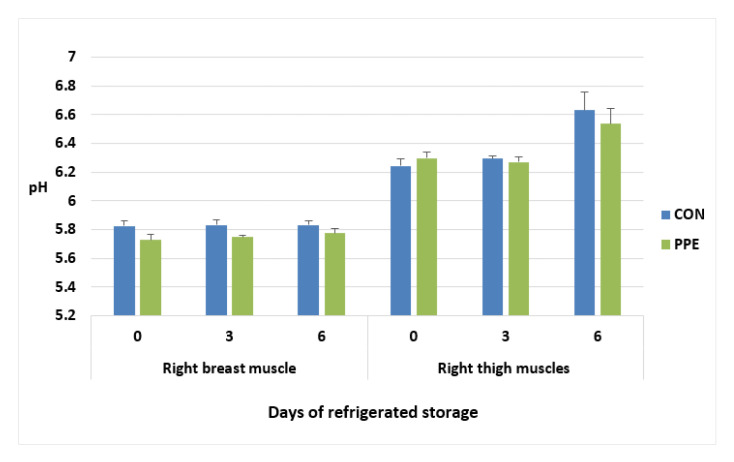
pH values of breast (*pectoralis major*) muscle and thigh of Hubbard chickens fed control diet (CON) and diet supplemented with plant extract mixture (PPE) in relation to refrigerated storage time. n = 10; data are reported as mean ± SEM. Breast muscle: Treatment *p* = 0.062; Time *p* = 0.431; Treatment x Time *p* = 0.591. Thigh: Treatment *p* = 0.738; Time *p* = 0.001; Treatment x Time *p* = 0.431.

**Figure 2 animals-12-02599-f002:**
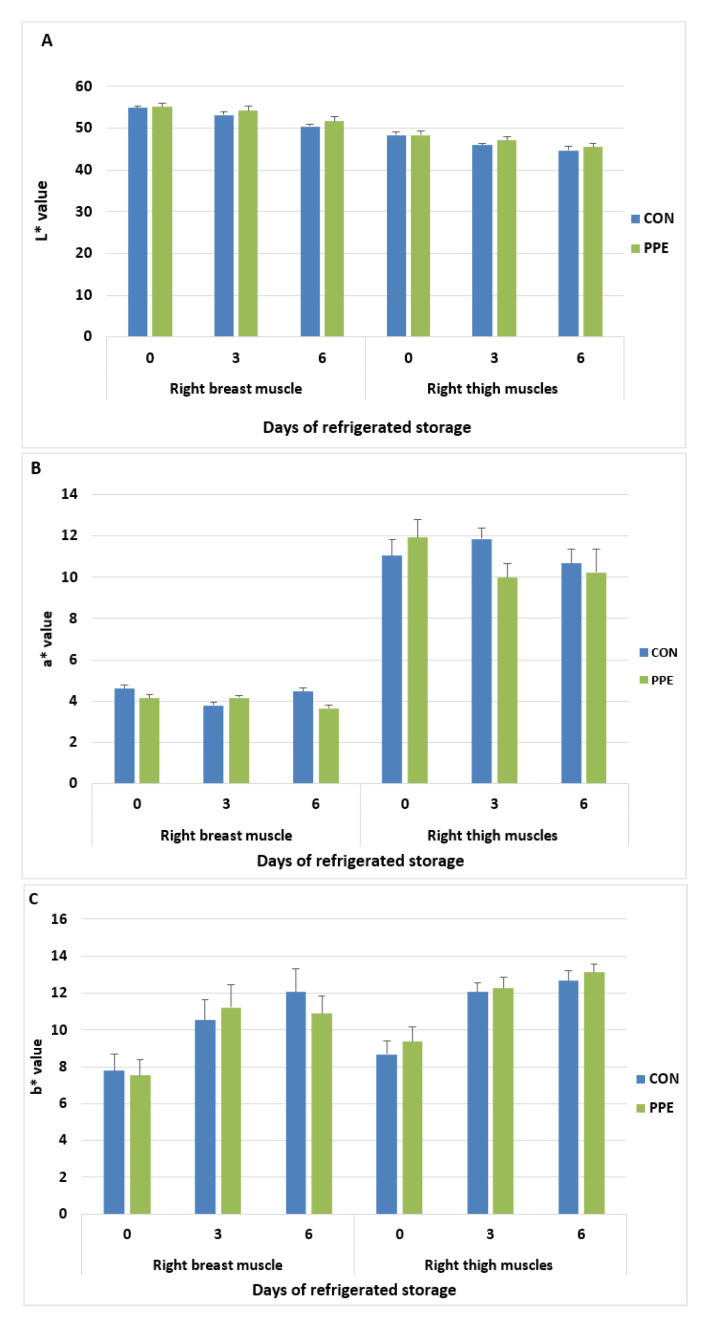
Colour indices, lightness (L*) (**A**), redness (a*) (**B**) and yellowness (b*) (**C**) of breast muscle (*m. pectoralis major*) and thigh of Hubbard chickens fed control diet (CON) and diet supplemented with plant extract mixture (PPE) in relation to refrigerated storage time. n = 10; data are reported as mean ± SEM. Breast muscle: Lightness (L*) values: effects of treatment, *p* = 0.297; time, *p* < 0.001; time ∗ treatment, *p* = 0.475; Redness (a*) values: effects of treatment, *p* = 0.498; time, *p* = 0.298; time ∗ treatment, *p* = 0.160; Yellowness (b*) values: effects of treatment, *p* = 0.841; time, *p* < 0.001; time ∗ treatment, *p* = 0.429. Thigh: Lightness (L*) values: effects of treatment, *p* = 0.291; time, *p* < 0.001; time ∗ treatment, *p* = 0.258; Redness (a*) values: effects of treatment, *p* = 0.567; time, *p* = 0.344; time ∗ treatment, *p* = 0.156; Yellowness (b*) values: effects of treatment, *p* = 0.539; time, *p* < 0.001; time ∗ treatment, *p* = 0.878.

**Figure 3 animals-12-02599-f003:**
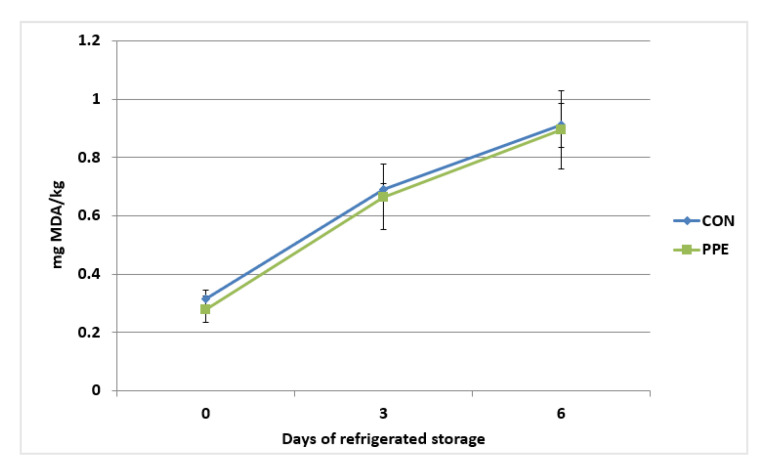
Oxidative stability of the breast (*pectoralis major*) muscle of Hubbard chickens fed control diet (CON) and diet supplemented with plant extract mixture (PPE) in relation to storage time. n = 10; data are reported as mean ± SEM. Effects of treatment, *p* = 0.738; time, *p* < 0.001; time ∗ treatment, *p* = 0.980.

**Figure 4 animals-12-02599-f004:**
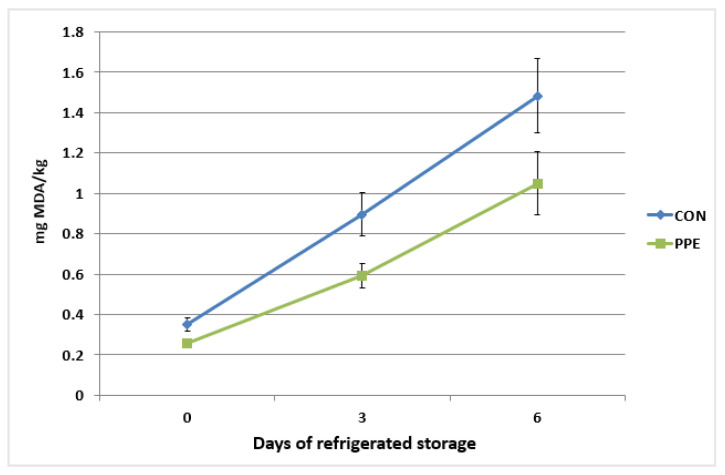
Oxidative stability of thigh muscle of Hubbard chickens fed control diet (CON) and diet and diet supplemented with plant extract mixture (PPE) in relation to storage time. n = 10; data are reported as mean ± SEM. Effects of treatment, *p* = 0.016; time, *p* < 0.001; time ∗ treatment, *p* = 0.293.

**Table 1 animals-12-02599-t001:** Chemical composition and polyphenols content of the dietary supplement.

Item	
	% DM
Dry matter	93.6 ± 5.05
Crude protein	7.2 ± 0.99
Ether extract	0.32 ± 0.01
Crude fiber	11.2 ± 1.02
Carbohydrates	49.6 ± 3.18
Ash	32.7 ± 1.38
Chemical compounds: ^a^	mg/kg DM
β-Carotene	402 ± 30.89
Phenolic acid:	
Syringic acid	1059.8 ± 62.82
Hydroxycinnamic acids:	
Neochlorogenic acid	7979.2 ± 468.11
Rosmarinic acid	126.5 ± 8.67
Trans-sinapic acid	105.5 ± 8.09
Chlorogenic acid	21.4 ± 3.65
Tannins:	
Ellagic acid	2440.9 ± 148.29
Rutin	272.4 ± 20.82
Flavonoids:	
Myricetin	53.9 ± 5.68

^a^ Values are expressed as means (n = 4) ± standard deviation.

**Table 2 animals-12-02599-t002:** Chemical composition of breast (*pectoralis major*) muscle of Hubbard chickens fed control diet (CON) and diet supplemented with plant extract mixture (PPE).

Item ^1^	CON	PPE	*p* Value
Moisture %	72.13 ± 0.11	71.97 ± 0.14	0.372
Crude Protein, % ^2^	23.02 ± 0.68 ^a^	24.80 ± 0.15 ^b^	0.025
Crude fat, % ^2^	0.92 ± 0.06	0.97 ± 0.16	0.490
Ash, % ^2^	1.15 ± 0.02	1.27 ± 0.05	0.137

^1^ Data are reported as mean values ± SEM, n = 10. ^2^ Data expressed as percentage of wet weight. ^a^, ^b^ on the same row differed for *p* < 0.05.

**Table 3 animals-12-02599-t003:** Chemical composition of thigh muscle of Hubbard chickens fed control diet (CON) and diet supplemented with plant extract mixture (PPE).

Item ^1^	CON	PPE	*p* Value
Moisture %	73.99 ± 0.26	73.63 ± 0.20	0.280
Crude Protein, % ^2^	21.18 ± 0.36	21.29 ± 0.34	0.821
Crude fat, % ^2^	3.10 ± 0.26	3.41 ± 0.31	0.474
Ash, % ^2^	1.01 ± 0.02	1.03 ± 0.01	0.528

^1^ Data are reported as mean values ± SEM, n = 10. ^2^ Data expressed as percentage of wet weight.

## Data Availability

The data presented in this study are available on request from the corresponding author.

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
