# Peer review of "Poultry Meat Quality in Antibiotic Free Production Has Improved by Natural Extract Supplement"

_animals, 2022, doi:10.3390/ani12192599_

Round 1
Reviewer 1 Report
It is a very interesting topic included in the paper. It is suggested to expand in Discusion the positive mechanisms attributable to the supplement

Author Response
Authors would like to thank the Reviewer 1 for the positive comment on our submitted manuscript.
The required corrections are reported below:
L42 : changed into “identified” (new line 43)
L289: changed into “observed” (new line 300)
L309: deleted the repetition of “in” (new line 320)

Reviewer 2 Report
This manuscript describes the growth performance (to underestimated) and meat quality of female slow growing broilers with an antibiotic free diet supplemented with seaweed and polyphenols on performance and meat quality.
The design clearly does not consider comparisons with antibiotic supplements. That’s good. The effect of PPE on growth performance was not considered (why)? The evaluation of meat quality was well done but considered only female animals. This is not sufficient for a good description of this scientific research. The manuscript needs more relevant data.
In detail: English must be improved: Often present and past as well as singular and plural are not correctly used.
L. 24: What was the origin of the supplements?
L. 86: Material and methods
L. 91 ff: Give more details. What was the size of the cages etc., why only female animals? etc.,
L. 98 ff: How were the feeds composed, what plants were used for plant extracts, how was the extraction, etc.
L. 105 ff: Give composition and nutrient content of the diets, use Joule not kcal.
Tab. 1: Analyzed values?
L. 167 ff: Add a Tab. with data on BW, DBWG, FI, FCR, mortality etc., what was n?, What was done with all other animals? Comment on these data in the discussion. Can such low growth performances be compared with other data?
Fig.1 – 5: Make a new Tab. With all these data. That’s far more informative.
Author Response
This manuscript describes the growth performance (to underestimated) and meat quality of female slow growing broilers with an antibiotic free diet supplemented with seaweed and polyphenols on performance and meat quality. The design clearly does not consider comparisons with antibiotic supplements. That’s good. The effect of PPE on growth performance was not considered (why)? The evaluation of meat quality was well done but considered only female animals. This is not sufficient for a good description of this scientific research. The manuscript needs more relevant data.
Authors thank the Reviewer for the comment. The growth performance data in relation to dietary treatments are reported in the manuscript. The author decided to not include the term “growth performances” only in the title. Now the term “growth performances” is included in the keywords. Data are analyzed and reported in the result and discussion sections. In the present research the authors considered only female animals because in field condition, the production system for broiler chickens in Italy is mainly characterized by the separate farming of females and males.
In detail: English must be improved: Often present and past as well as singular and plural are not correctly used.
English form has been revised.
L 24: What was the origin of the supplements?
In the abstract “natural extract” has been added. (new line 24)
- 86: Material and methods
The mistake has been corrected (new line 88)
- 91: Give more details. What was the size of the cages etc., why only female animals?
Detailed about stocking density has been now reported in the text (new line 95-96). The authors considered only female animals because in field condition, the production system for broiler chickens in Italy is mainly characterized by the separate farming of females and males.
- 98: How were the feeds composed, what plants were used for plant extracts, how was the extraction, etc.
Feed composition and plant extract used have been added in the text. (new line 105 and 110)
- 105: Give composition and nutrient content of the diets, use Joule not kcal.
As suggested, the energy content of the diet has been reported in MJ/kg (new line 113-114)
Tab. 1: Analyzed values?
The analytical methods for chemical composition has been now reported in the text. (new line 108-109)
- L. 167 ff: Add a Tab. with data on BW, DBWG, FI, FCR, mortality etc., what was n?, What was done with all other animals? Comment on these data in the discussion. Can such low growth performances be compared with other data?
Authors thank the Reviewer for the suggestion but prefer to present the all the data on growth performance in the text (new line 296-314), instead of a table, since the manuscript already contains 8 graphical representations (3 tables and 5 figures) and a new table on growth performance seems to be redundant. Data on mortality have been reported. (new line 169-170 and 184-185).
Regarding the number of repetitions in new Line 173-174 is reported that “Pen was considered as experimental unit for growth performances.”, so n=10.
The performance parameters in relation to dietary treatments were already discussed in the discussion section reporting a sufficient number of citations for data comparison. (new Line L296 to L314) The other animal had a market destination.
Fig.1 – 5: Make a new Tab. With all these data. That’s far more informative.
The authors thank the Reviewer for the suggestion. We prefer to maintain the graphical representation of the data because the trend of the data, related to sampling time, are more comprehensive and readily understandable.

Reviewer 3 Report
The purpose of this study was to determine the effects of dietary supplementation with brown seaweeds and plant polyphenols mixture on growth performance, proximate chemical composition, meat colour parameters, and meat oxidative stability of slow growing Hubbard chickens. The Introduction chapter provides an overview of the world's knowledge on this subject. The material used in the research is sufficiently numerous, but some supplementing the description in Materials and Methods chapter are needed. The results are described correctly. The discussion is exhaustive. Summary of the results are correct. Some corrections are needed before publishing an article in Animals. The proposed changes are listed below.
General comments:
Please prepare the article in accordance with the instructions for authors:
In Affiliation, enter the initials of the name and surname of each co-author of the article
On the 1st page on the left side, add "Citaion" and the required data
For significance please use lowercase "p" in italic instead of uppercase "P" throughout the main article
Detailed comments:
L14 antibiotic free production - is it also therapeutically free from antibiotics? Is it only from atibiotic growth promoters (AGP)
L25 do you know the full name of the hybrid (Hubbard JA57? or Hubbard JA 57 Ki? or Hubbard JA87?)
L28 experimental group is the PPE group?
L29 Dietary supplementation PPE did not….
L62 Please write something that ... .. chicken meat is also tender, easily digestible, easy to cook, cheaper than pork and beef and meat of other poultry, which increased its importance in the human diet (see Kokoszyński et al. 2022 in Animal Science Journal)
L94 add pen size information
L95 and others before the "°C" add space
L985 provide information on light length, intensity, type and color
L116 provide the name of the balance, manufacturer's data, measurement accuracy for initial and finish BW, feed intake measurement
L124 right? thigh
L130 How many minutes / hours after slaughter were the determinations for the term "0" day
L133 how technically did the pH measurement, depth and angle determinations?
L135 enter data for Y =, x =, y = for white reference tile
L165 „Individual bird (chicken)” instead of current form
L169 and others (p > 0.05) or (p < 0.05) instead of current forms
L170 add measurement units (g / day) after „37.30 ± 3.4 PPE”
L177 "Figure" with a capital letter
Figure 1 and Figure 2A and 2C "Right breast muscle" instead of Right breast; no significance markings in the chart
L231 breast muscle not thigh muscle
In Table 2 and 3: ± "space" before the SEM value; after the table title and explanation dot; "p-Value" instead of P value; 23.02 ± 0.68b; 24.80 0.15a; from the above value of the "a"
L25 0 3.22% or 3.25%?
L311 delete one "a dot" after treatment
L352 "physical" parameters or „physicochemical”
L390 "Ital. J. Anim. Sci. " instead of current form
L425 "Ital. J. Anim. Sci. " instead of current form
Author Response
The purpose of this study was to determine the effects of dietary supplementation with brown seaweeds and plant polyphenols mixture on growth performance, proximate chemical composition, meat colour parameters, and meat oxidative stability of slow growing Hubbard chickens. The Introduction chapter provides an overview of the world's knowledge on this subject. The material used in the research is sufficiently numerous, but some supplementing the description in Materials and Methods chapter are needed. The results are described correctly. The discussion is exhaustive. Summary of the results are correct. Some corrections are needed before publishing an article in Animals. The proposed changes are listed below.
General comments:
Please prepare the article in accordance with the instructions for authors:
In Affiliation, enter the initials of the name and surname of each co-author of the article
On the 1st page on the left side, add "Citaion" and the required data
For significance please use lowercase "p" in italic instead of uppercase "P" throughout the main article
Authors thank the Reviewer 3 for the positive comment. All the above suggestions have been addressed in the revised manuscript.
Detailed comments:
L14 antibiotic free production - is it also therapeutically free from antibiotics? Is it only from atibiotic growth promoters (AGP)
Authors confirm that the experimental trial was conducted with free antibiotic production, both therapeutic and growth promoter procedure.
L25: do you know the full name of the hybrid (Hubbard JA57? or Hubbard JA 57 Ki? or Hubbard JA87?)
The full name of the hybrid has now reported in the text. (Hubbard JA57) (new line 24 and 94).
L28: experimental group is the PPE group?
The acronymous for the experimental group (PPE) has been added in the text (new line 29).
L29: Dietary supplementation PPE did not….
The suggested change has been made. (new line 30)
L62: Please write something that ... .. chicken meat is also tender, easily digestible, easy to cook, cheaper than pork and beef and meat of other poultry, which increased its importance in the human diet (see Kokoszyński et al. 2022 in Animal Science Journal)
Authors thank the Reviewer for the suggestion. These claims are already mentioned in the reported citation (Lusk, J.L. Consumer preferences for and beliefs about slow growth chicken. Poult. Sci. 2018). Authors added these claims in the text. (new line 64-66)
L94: add pen size information
The requested information has been added in the text. (new line 95-96)
L98: and others before the "°C" add space
The suggested change has been made.
L98: provide information on light length, intensity, type and color
Authors provided details on light regimen as required. (new line 98)
L116 provide the name of the balance, manufacturer's data, measurement accuracy for initial and finish BW, feed intake measurement
The data has been reported in the text. (new line 120-121)
L124: right? Thigh
The mistake has been corrected. (new line 130)
L130: How many minutes / hours after slaughter were the determinations for the term "0" day
Authors confirm that the “0 day” means between 4-5 hours from the slaughter.
L133: how technically did the pH measurement, depth and angle determinations?
The pH value of the breast muscle was performed by inserting a probe electrode into the cranial ventral part of the muscle. In the thigh muscle the pH value was performed by inserting a probe electrode at the level of biceps femoris muscle. (new line 136-142)
L135 enter data for Y =, x =, y = for white reference tile.
Usually this information is not reported in the text, because the white calibration plate was included by manufacturer (Calibration Plate CR-A43; Minolta Camera Co.) and have an Y=91.59; x=0.3147 and y=0.3311
L165: „Individual bird (chicken)” instead of current form
The suggested change has been made. (new line 174)
L169 and others (p > 0.05) or (p < 0.05) instead of current forms
The suggested change has been made.
L170: add measurement units (g / day) after „37.30 ± 3.4 PPE”
The suggested change has been made. (new line 181)
L177 "Figure" with a capital letter
The suggested change has been made. (new line 189)
Figure 1 and Figure 2A and 2C "Right breast muscle" instead of Right breast; no significance markings in the chart
The suggested change has been made. The significance is reported in the caption.
L231: breast muscle not thigh muscle
The suggested change has been made. (new line 241)
In Table 2 and 3: ± "space" before the SEM value; after the table title and explanation dot; "p-Value" instead of P value; 23.02 ± 0.68b; 24.80 0.15a; from the above value of the "a"
The suggested change has been made
L258:….0 3.22% or 3.25%?
The mistake has been corrected. (new line 261)
L311:…. delete one "a dot" after treatment
The suggested change has been made.
L352:…. "physical" parameters or „physicochemical”
The suggested change has been made. (new line 363)
L390: …. "Ital. J. Anim. Sci. " instead of current form
The suggested change has been made. (new line 401)
L425:…."Ital. J. Anim. Sci. " instead of current form
The suggested change has been made. (new line 437)
